# Recruitment and retention interventions in surgical and wound care trials: A systematic review

Catherine Arundel[ID]*, Andrew Mott[ID]

York Trials Unit, Department of Health Sciences, University of York, Heslington, York, United Kingdom

* catherine.arundel@york.ac.uk

## Abstract

### Background

Recruitment and retention to surgical trials has previously been reported to be problematic, resulting in research waste. Surgery often results in wounds, meaning these trials are likely to have similar populations. There is currently no systematic assessment of effective strategies for these populations and hence, systematic assessment of these was deemed to be of importance.

### Methods

A systematic review was conducted. Studies were eligible if they were randomised controlled trials undertaken to test an intervention to improve recruitment or retention within a surgical or wound based host randomised controlled trial. MEDLINE, EMBASE, Cochrane Library, ORRCA Database and the Northern Ireland Hub for Trials Methodology Research SWAT Repository Store were searched. Two independent reviewers screened the search results and extracted data for eligible studies using a piloted extraction form. A narrative synthesis was used due to a lack of heterogeneity between strategies which prevented meta-analysis.

### Results

A total of 2133 records were identified which resulted in 13 ultimately being included in the review; seven on recruitment and six on retention. All included studies were based within surgical host trials. Four of the seven recruitment studies focussed on the provision of consent information to participants, one focussed on study set up and one on staff training, with only one relating to consent information finding any significant effect. A range of retention strategies were assessed by the included studies, however only two found (pen vs no pen, mailing strategies) found any significant effect.

### Conclusion

The included studies within a trial were all conducted within surgical trials. There was significant variation in strategies used, and limited replications and therefore further assessment

**Data Availability Statement:** Datasets and associated documentation used in this study will be available in the FigShare database https://figshare.com/ 10.6084/m9.figshare.23580270.

**Funding:** The author(s) received no specific funding for this work.

**Competing interests:** The authors have declared that no competing interests exist.

may be warranted. Given the lack of studies embedded within wound care trials, further studies in this area are recommended.

## Trial registration

PROSPERO (CRD42020205475).

## Introduction

Fundamental to health research is the testing of interventions through randomised controlled trials (RCTs). The validity and reliability of RCTs is highly dependent on recruiting and retaining sufficient numbers of participants [1]. Reviews [2–5] have shown that RCTs have consistently struggled with recruitment and this continues to prevail with the most recent review demonstrating that only 63% of RCTs reviewed achieved the required sample size [2]. Approximately a quarter of trials also experience attrition resulting in greater than 10% of primary outcome data being unavailable for use in the end analysis [6]. Limited recruitment and retention can result in a number of issues, for example additional costs, the need for a study extension, reduced power and early termination of research activity, therefore resulting in significant research waste [1, 7].

Many methods to improve recruitment and retention are utilised by trialists, often with limited robust evidence to support their effectiveness. As a result, evidence-based methods to increase recruitment and retention to RCTs are becoming extremely necessary and valuable. The most robust way to assess recruitment and retention interventions is to embed or nest a randomised evaluation within a host trial, known as a study within a trial or SWAT [8]. Testing interventions in such a way ensures causality of intervention effectiveness is assessed [9].

The testing of recruitment and retention strategies has increased in recent years and strategies have been combined in Cochrane Reviews [10, 11]. Only a small number of interventions have provided strong evidence of their potential to affect recruitment [11] and there is currently no high certainty GRADE evidence for retention strategies [10]. In addition, in some instances evidence of effectiveness included in these reviews is derived from quasi, hypothetical, or non-randomised SWAT designs which may limit the applicability of effectiveness findings to a RCT design.

There is evidence that both the recruitment and retention rate of trials are strongly linked to the setting in which they are undertaken [3]. Although many strategies may be transferable across clinical populations and study settings, there may also be unique characteristics which make specific interventions more, or less, effective in certain settings. Despite this, the evidence for effectiveness of strategies for specific groups (clinical populations, research settings) remains limited.

Over 10 million surgical operations take place within the UK NHS on an annual basis [12] and approximately 2.2 million patients will have a chronic wound at any one time [13]. As a result, a significant number of surgery or wound care research studies will be ongoing at any one time.

It has been identified that one in five surgical trials are discontinued due to lack of recruitment, which is a huge source of research waste [14]. Reasons for this include clinician and patient treatment preferences, overestimation of the eligible patient pool and clinician time constraints (Crocker et al. [15]). Similarly, retention in surgical trials has also previously been reported to be problematic, particularly due to patient dissatisfaction in not receiving their

preferred treatment, when no treatment is required after the initial procedure, or where a long follow up period is used [16, 17].

Surgical procedures often lead to wounds and so trials of surgical or wound care are likely to share similar populations. These populations may differ from those in other forms of trial, due to the nature and trajectory of the associated interventions and follow up, and so may respond differently to strategies tested. To our knowledge, no systematic assessment has been made of the effectiveness of recruitment and retention strategies for these populations. Given the ongoing surgical and wound care trials at our UKCRC registered clinical trials unit, and the limited evidence for effective strategies in specific groups, it was viewed that assessment of effective strategies for this sector was of importance [18].

This review therefore sought to establish the evidence base for strategies to improve the recruitment and retention of patients to surgical and wound care clinical trials. The secondary aims of this review were to identify gaps in the evidence base for RCTs in these patient populations and to evaluate the cost effectiveness of different strategies (cost per patient recruited or retained) for any interventions shown to be effective.

## Methods

### Protocol

A protocol for this systematic review was prospectively registered on PROSPERO (CRD42020205475) on the 22nd October 2020.

### Eligibility criteria

Studies were eligible for inclusion if they

- Enrolled adult participants (≥18 years) into a surgical or wound care randomised controlled trial (commonly referred to as the host trial).

- Used a randomised controlled trial to test an intervention to improve either recruitment or retention to the surgical or wound trial (commonly referred to as a Study within a Trial or SWAT)

Studies were not eligible if either the SWAT or host trial was hypothetical, quasi-randomised or non-randomised.

### Information sources and search strategy

Using previously published search strategies for recruitment and retention strategies in other patient groups, a search strategy was designed to identify published randomised trials which focussed on improving recruitment and retention in surgical and wound care randomised trials.

The strategy included three core components: recruitment or retention; randomised controlled trials; surgery and wound. The only limitation applied was that the articles were published in English. A copy of the full search strategy is included as S1 File.

Electronic databases including MEDLINE, EMBASE, Cochrane Library, ORRCA Database and the Northern Ireland Hub for Trials Methodology Research SWAT Repository Store were searched from date of inception to the date of the search on 26th January 2021. A further search to MEDLINE and EMBASE was undertaken on 7th February 2022 to identify any publications since the initial search.

In addition, article reference lists and bibliographic searches were undertaken during the screening process. The PROMETHEUS programme [19, 20] (hosted by York Trials Unit,

University of York) was also contacted to obtain an update on the progress of any relevant SWATs.

### Selection process

Titles and abstracts retrieved from the searches were downloaded into Rayyan (https://www.rayyan.ai/) and de-duplicated. The remaining titles and abstracts were independently screened by two reviewers (CA and AM) against the pre-specified inclusion and exclusion criteria. Full text copies were obtained for those articles deemed to be meeting inclusion criteria and these were again independently reviewed by two reviewers (CA and AM). Where necessary, documentation relating to the host trial (for example registry entry, protocols, published results) were obtained to aid eligibility assessment. In both instances, any disagreements were discussed and resolved.

### Risk of bias assessment

The Cochrane Risk of Bias tool (version 2) was used to assess risk of bias [21], applying all domains of the tool. An assessment was made only of the SWAT outcomes and not of the host trials. Two reviewers independently assessed the risk of bias for each included outcome and any disagreements in assessment were resolved by discussion.

### Confidence in cumulative evidence

The strength of the evidence was assessed using GRADE [22]. An assessment was made only of the SWATs and not of the host trials. One reviewer independently assessed GRADE for each included study and this assessment was reviewed and agreed with the second reviewer.

### Data collection and items

Using a standardised data extraction form, data extraction was completed independently by two reviewers (CA and AM) and compared for consistency. The extraction form was piloted prior to full data extraction.

### Outcome data

Outcome data were collected on the number of participants either recruited (i.e., consented and randomised) or retained (i.e., providing outcome data) within each SWAT at any time point.

Secondary outcomes collected included:

- Cost-effectiveness: defined as cost per additional participant recruited or retained.

- Additionally for retention SWATS the retention of participants at subsequent timepoints was assessed.

### Data items

Data was collected regarding the characteristics of both the host trial and the SWAT. The following items were collected:

Host trial:

- Clinical Specialty

- Surgical or Wound Care trial

- Setting (primary or secondary care)

- Trial Design

- Trial Interventions

- Total required sample size

- Primary Outcome Measure

- Recruitment method (remote, in clinic, etc.)

- Follow-up methods (remote, in clinic, etc.)

  SWAT:

- Trial Design

- Participant characteristics

- Intervention details

- Comparator details

- Number of participants or sites recruited to each arm

- Primary outcome

- Number of participants recruited or retained

- Secondary Outcomes collected

- Cost effectiveness

- Number of participants recruited or retained at further timepoints

### Synthesis

A study flowchart of the study selection process is presented. Key study characteristics are summarised in tables and trials will be grouped by type of intervention.

A narrative synthesis is presented for each intervention. Studies at high risk of bias will be included in the results however all results will be discussed within the context of the ROB assessment. Where available data of the cost effectiveness of an intervention will be presented if an intervention has been shown to be effective. No sub-group or sensitivity analyses were planned.

Data from studies with multiple publications were extracted and reported as a single study. Multiple recruitment or retention interventions tested within the same host trial were extracted and treated as separate studies.

## Results

In total our searches identified 2189 records of which 70 were identified as duplicates. Of the 2119 screened, 25 were included for full-text review. Following full-text review, 12 records were eligible for inclusion [23–34]. A further record was subsequently identified for inclusion on the basis that 62.5% of studies included in the record were surgical or wound care related [35]. This resulted in a total of 13 included records [23–35].

A study flowchart is presented in Fig 1. Five additional studies were identified that were ongoing [36–39].

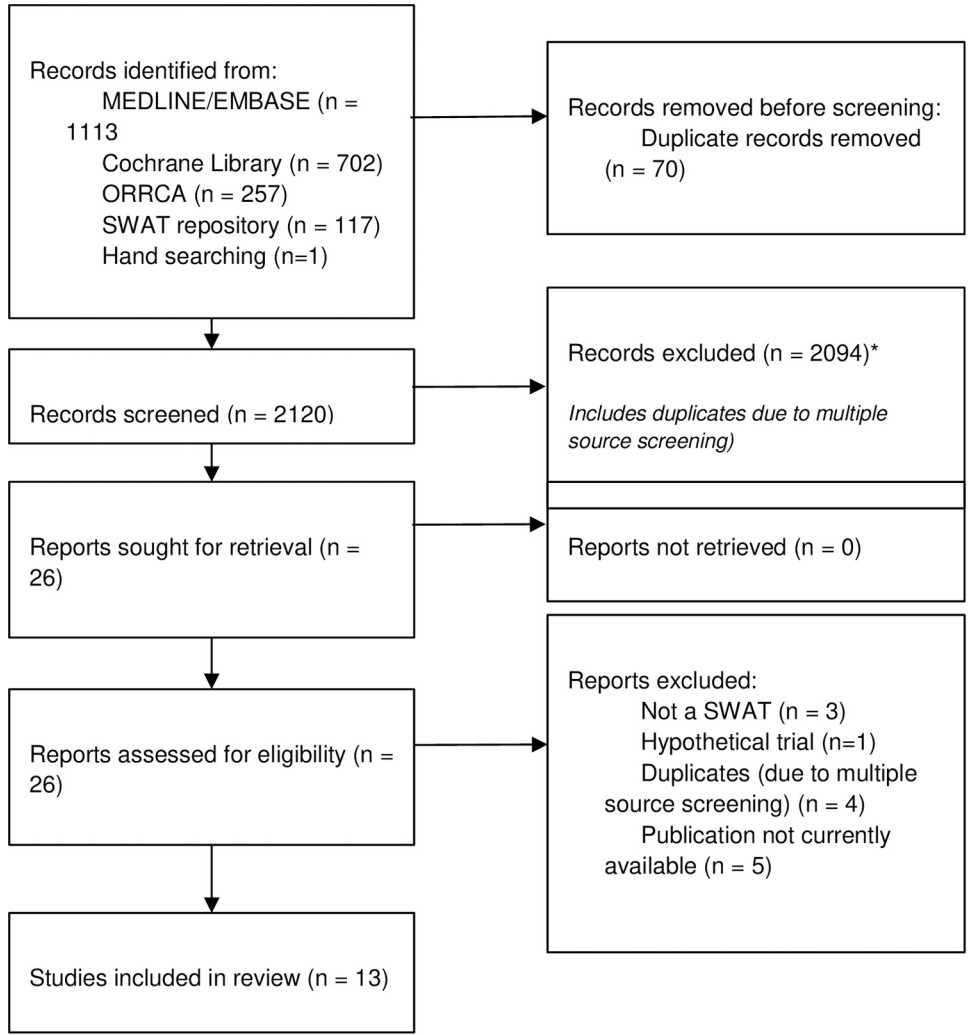

**Fig 1. PRISMA flow diagram of screening.**

## Study characteristics

Of the studies included seven tested interventions addressing recruitment and six tested interventions addressing retention. Additionally, five ongoing studies were identified. The studies were primarily surgical, with two being wound-based. The recruitment method for all studies was a direct approach (i.e., face to face recruitment). The retention method for the majority of SWATs addressing retention was postal questionnaire follow-up with two studies included by Coleman et al. [35], using a combination of postal and clinic follow up (Table 1).

## Risk of bias

The outcomes of the 13 studies included in this review were assessed using the Cochrane Risk of Bias 2 tool, and results are summarised in Fig 2.

Overall the included studies were reasonably well reported with eight studies with low risk of bias [26, 28–30, 32–34], four with some concerns with bias [23, 24, 27, 31] and one study assessed as being at high risk of bias [25]. Donovan et al. was considered high risk of bias due

**Table 1. Study characteristics.**

| | Author (year) | Clinical Speciality | Surgical/ Wound | Host Recruitment method | Host Follow-up Method | Planned Sample Size (Host: SWAT) | SWAT Intervention | SWAT Control | Outcomes Collected |
|---|---|---|---|---|---|---|---|---|---|
| Recruitment | Abd-Elsayed et al. (2012) [23] | Cardiology | Surgical | Direct approach | NS | NS:526 | Enhanced consent documents | Standard consent documents | Proportion consenting to trial |
| | Brubaker et al. (2019) [24] | Obstetrics & Gynacology | Surgical | Direct approach | Clinic Visit | 374:340 | Information video | Standard consent | Proportion consenting to trial Proportion completing extended follow-up |
| | Eccles et al. (2002) [32] | Urology | Surgical | Direct approach | NS | 400:30 | Decision Aid Video | Standard consent process | Proportion of participants randomised to trial |
| | Donovan et al. (2003) [25] | Urology | Surgical | Direct approach | NS | NS:150 | Nurse Provided information | Urologist provided information | Number recruited to trial |
| | Jefferson et al. (2018) [26] | Orthopaedics | Surgical | Direct approach | Postal or Clinic visit | 438 | In person Study Set up | Remote Study set up | Time to: R&D approval final site initiation visit first randomised participant number of participants screened proportion of eligible participants randomised |
| | Parker et al. (2022) [33] | Multiple Specialities (SWAT covering four trials) | Surgical | Direct Approach | Postal follow-up | NA | Study Site receives QuinteT Recruitment Intervention & GRANULE online training | No Training | 1) Feasibility & acceptability of intervention. 2) Participant screening and recruitment rate (defined as the proportion of eligible participants who gave their consent and were randomised into the host trial six months following delivery of the course). |
| | Agni et al. (2022) [34] | Orthopaedic | Surgical | Direct Approach | Telephone or postal | 4106:NS | Enhanced Trainee Principal Investigator (TPI) package; Digital Nudge; TPI and Digital Nudge | Usual Practice | Proportion of participant randomised (in first 6months of recruitment) |

*(Continued)*

**Table 1.** (Continued)

| | Author (year) | Clinical Speciality | Surgical/ Wound | Host Recruitment method | Host Follow-up Method | Planned Sample Size (Host: SWAT) | SWAT Intervention | SWAT Control | Outcomes Collected |
|---|---|---|---|---|---|---|---|---|---|
| **Retention** | Watson A et al. (2017) [27] | Gastroenterology | Surgical | Direct approach | Postal Follow-up | 800:600 | Vouchers at one or two follow-up time points (12 & 24 month) | No Voucher | Response rate at each follow-up timepoint |
| | Mitchell et al. (2020a) [29] | Orthopaedics | Surgical | Direct approach | Postal Follow-up | 2600:2306 | Inclusion of pen with follow-up | No pen | Response rate at follow-up |
| | Mitchell et al. (2020b) [29] | Orthopaedics | Surgical | Direct approach | Postal Follow-up | 2600:1470 | Personalised SMS reminder of follow-up | Standard SMS reminder of follow-up | Response rate at follow-up |
| | Sarathay et al. (2020) [30] | Orthopaedics | Surgical | Direct approach | Postal Follow-up | 500:269 | pre-notification SMS of questionnaire | SMS after questionnaire posted | Response rate to follow-up |
| | Coleman et al. (2021) [35] | Surgical: (Orthopaedic n = 2; Urology n = 1; Gastroenterology n = 1) Wound care: Vascular n = 1 | Surgical/ Wound | NS | Postal Follow up or Postal/ Clinical follow up | NS | Festive greetings card | No festive greetings card | Response rate at follow up |
| | Renfroe et al. (2002) [31] | Cardiology | Surgical | Direct approach | Clinical Review & postal follow-up | 1200:664 | Express delivery of questionnaire | Regular Mail | Response rate to follow-up |
| | | | | | | | Certificate of Appreciation | No certificate | |
| | | | | | | | Early delivery (1–2 weeks) | Later delivery (1–4 months) | |
| | | | | | | | Study coordinator signed letter | PI signed letter | |
| **Unpublished / Ongoing** | Reed et al. [39] | Orthopaedics | Surgical | Direct Approach | Based on SWAT | Ongoing | Postal Follow-up | Telephone Follow-up | Response rate to follow-up |
| | Starr et al. [36] | Urology | Surgical | Direct approach | Paper Questionnaires provided following intervention | Ongoing | Theoretically informed leaflet in the participant pack | Generic compliments slip in the participant pack | Response rate at follow-up |
| | Arundel et al. [38] | Vascular | Wound | Direct Approach | Clinical review & postal follow-up | Ongoing | Sending of Thank You card between follow-ups | Usual Follow-up | Proportion of participants returning questionnaire at first postal follow-up. |
| | McCaffery et al. [37] | Vascular | Wound | Direct Approach | Clinical review & postal follow-up | Ongoing | Infographic + Patient information leaflet | Patient information leaflet alone | Difference in site recruitment rate |
| | Montgomery et al. [40] | Oncology | Surgical | Direct approach | Telephone or clinic visit | Ongoing | Pictorial aid at end of information sheet depicting randomisation and trial treatment arms | Standard participant information sheet | Proportion of participants randomised |

NS: Not Specified; NA: Not Applicable.

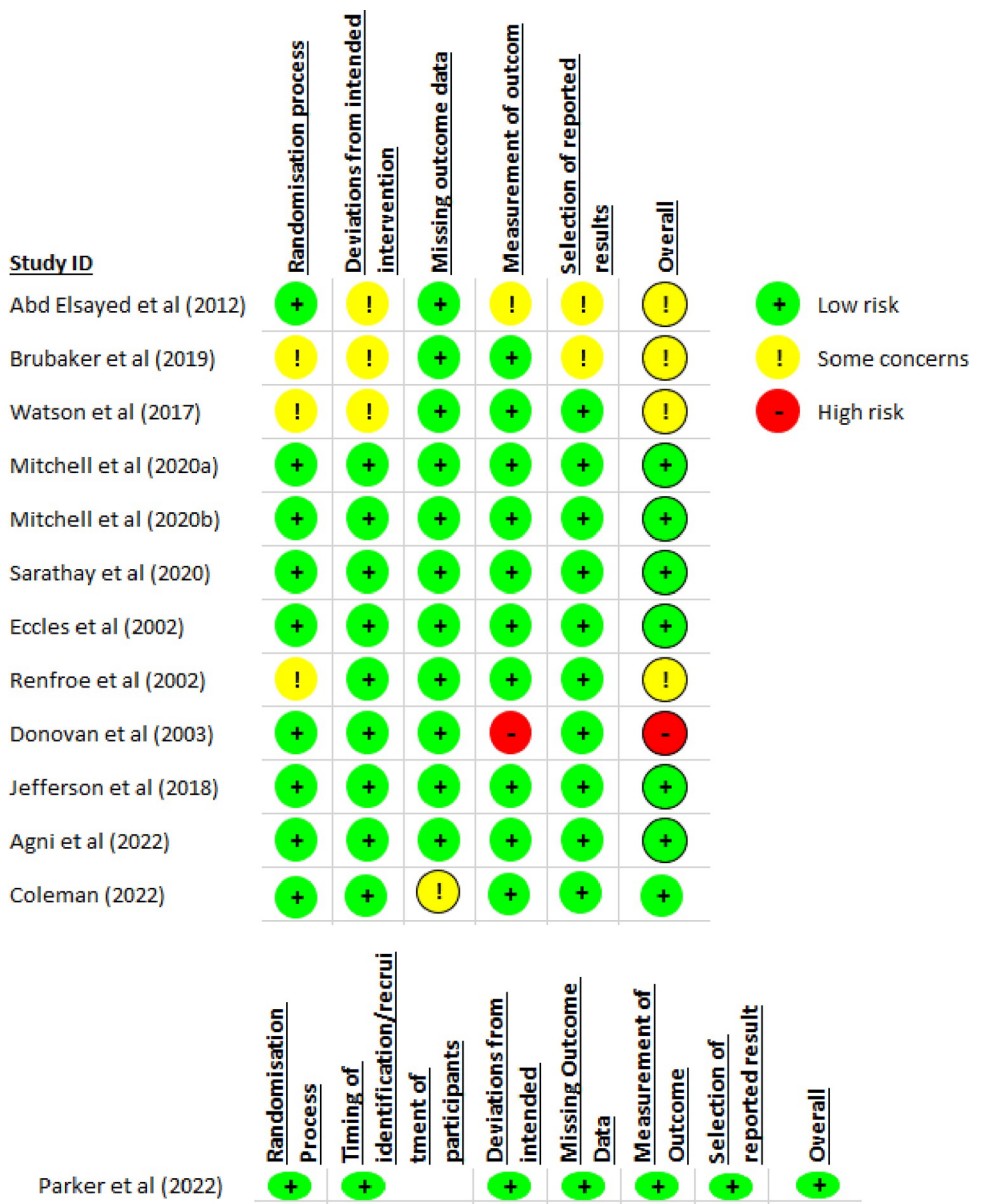

**Fig 2. Risk of bias assessment of included RCTs.**

to limited reporting in relation to measurement of outcomes. The reasons for some concerns were largely due to limited reporting of the randomisation process or deviations from intended intervention.

## GRADE assessment

Overall, certainty of evidence for the included studies, as assessed by GRADE, was deemed to be low. As detailed in S2 File, seven studies (64%) were deemed to have low GRADE assessment, three studies were deemed to have moderate GRADE evidence and two were deemed to have very low GRADE evidence. The main driver for the low GRADE evidence was associated risk of bias and/or imprecision arising due to wide confidence intervals or being a single study.

**Table 2. Summary of study results—Recruitment & retention outcomes.**

| Study | Total Participants | Control Group Total | Control group Recruited | Intervention Group Total | Intervention Group Recruited |
|---|---|---|---|---|---|
| Abd Elsayed et al. (2012) | 499 | 251 | 189 | 248 | 164 |
| Brubaker et al. (2019) | 305 | 152 | 143 | 153 | 142 |
| Donovan et al. (2003) | 150 | 75 | 53 | 75 | 50 |

| Study | Total Participants | Control Group Total | Control group Retained | Intervention Group Total | | Intervention Group Retained |
|---|---|---|---|---|---|---|
| Eccles et al. (2002) | 30 | 15 | 3 | 15 | | 1 |
| Jefferson et al. (2018) | 37 sites | 20 sites | N/A | 17 sites | | N/A |
| Parker et al. (2022) | NR | NR | NR | NR | | NR |
| Agni et al. (2022) | 20 sites | 6 sites | 379 participants | 5<br>4 sites<br>5 | | 279<br>147 participants<br>410 |
| Watson et al. (2017) | 521 | 132 | 12M: 98<br>24M: 98 | Voucher 12M | 142 | 12M:98<br>24M:100 |
| | | | | Voucher 24M | 123 | 12M:90<br>24M:86 |
| | | | | Voucher Both | 124 | 12M:93<br>24M:92 |
| Mitchell et al. (2020a) | 2306 | 1147 | 982 | 1146 | | 1020 |
| Mitchell et al. (2020b) | 1470 | 742 | 654 | 723 | | 644 |
| Renfroe et al. (2002)—Mail Type | 644 | 322 | 219 | 322 | | 242 |
| Renfroe et al. (2002) - Certificate | 644 | 322 | 242 | 322 | | 219 |
| Renfroe et al. (2002)– Timing | 644 | 322 | 232 | 322 | | 255 |
| Renfroe et al. (2002) - Letter signature | 644 | 322 | 226 | 322 | | 235 |
| Sarathay et al. (2020) | 269 | 134 | 119 | 135 | | 122 |
| Coleman et al. (2021) | 1103 (3223 including non-surgical/wound) | 547 | 289 | 553 | | 293 |

NR: Not Reported; M: Months.

## Analysis

Due to the lack of consistency in the interventions evaluated in the included studies it was inappropriate to undertake a meta-analysis and hence a narrative synthesis was conducted, grouped by recruitment and retention SWATs. A summary of the recruitment and retention outcomes is provided in Table 2.

## Recruitment

All seven recruitment SWATs identified were embedded within surgical host trials with a direct, face to face approach to participant recruitment. Only two SWATs [23, 24] reported participant demographic criteria, and in both instances the populations were older white caucasian adults (Abd-Elsayed: Age range 62 +/- 13 years, >90% caucasian; Brubaker: Age range 57, >80% white). The host trials included a range of conditions, with only two studies [25, 32] undertaken in the same area (Urology). All the included studies used individual randomisation across a range of surgical trials.

Four of the seven SWATs [23–25, 32] focussed on the provision of consent information to participants, one SWAT [26] focussed on study set up and two on staff training [33, 34].

All of the SWATs which focussed on consent information (modification to the consent process [18] and modification to how information was presented [19, 20, 27] reported a higher proportion of recruitment in the control arm compared with the intervention arm although there was no statistically significant difference in two of the studies [24, 25]. Abd-Elsayed [23] however reported that enhanced consent materials significantly reduced (p = 0.03) the odds of consenting.

The study by Jefferson et al. [26], compared an onsite face to face initial meeting (plus standard site initiation visit) with a remote initial meeting (plus standard site initiation visit) and identified that those sites who received the intervention had a higher consent rate compared to the control sites (0.63 vs 0.53), although the mean number of participants recruited favoured the control group (10 vs 11).

The study by Parker et al. [33] assessed the effect of a recruiter training course on obstacles and challenges to recruitment, derived from a synthesis of the QUINTET Recruitment Intervention [41], and online GRANULE training [42] on recruitment. The study identified no difference in the number of participants screened (coefficient −0.35, 95% CI -7.84 to 7.15, $p = 0.92$) or recruited (coefficient -0.07, 95% CI -0.43 to 0.29, $p = 0.66$) between sites that received the intervention and those that did not. The study by Agni et al. [34], assessed the effect of enhanced training and support for Trainee Principal Investigators (TPI) and personalised digital nudging to recruiters on recruitment rates. There was a statistically significant benefit to recruitment (Incidence rate ratio 1.23, 95% CI 1.09 to 1.40, p = 0.001) from the enhanced TPI intervention, but no significant effects were seen from the digital nudge component.

Cost effectiveness of the interventions was assessed only in two studies [25, 26]. The onsite face to face meeting was more costly than the remote initial meeting (£1016.93 vs £727.10) [26] and consent provision was cheaper when provided by a nurse vs a urologist (Difference 6.89, 95% CI 0.3 to 13.4, p = 0.039) [25].

A further two recruitment SWATs [37, 40] were identified during the search but remained ongoing, with no data reported, at the time of analysis. One of the studies was being hosted in a wound care trial [37] and one in a surgical trial [40]. Both focused on provision of information for participant consent.

## Retention

Of the six retention SWATs identified, the majority were embedded within surgical host trials with postal follow up. Renfroe et al. [31] also used clinical notes review as part of their follow up processes, and two of the included studies in Coleman et al. [35] used postal and clinical follow up, and included one wound care trial. The majority of the studies (n = 3, 60%) were hosted solely in orthopaedic surgical trials [28–30] with Coleman et al. also including two orthopaedic trials. Three of the SWATs were factorial [27, 31, 34], with the remaining studies using individual randomisation.

Participant demographics were well reported for the retention SWATs, with each publication providing age and gender, although ethnicity was only reported in two SWATs [31, 35]. Similarly, to the recruitment SWATs, participants were older (Range 49–76 years) however with a relatively even split between male and female participants (average 50.96% male). Two of the six SWATs [23–25, 32] focussed on the use of SMS (text messaging) with participants, with the remaining SWATs focussing on financial incentives [22], inclusion of pens [24], postal delivery methods [26] or festive greetings cards [35].

A range of retention interventions were assessed and none assessed the same intervention as any other, although Mitchell et al. [28] and Sarathay et al. [30] both used text messaging interventions (personalised reminder and prenotification respectively).

Only two SWATs identified statistically significant differences in retention. Mitchell et al. [29] found that including a pen with a postal questionnaire increased response rates by 3.4% (95% CI 0.7 to 6.1, p = 0.01) and Renfroe et al. [31] found that using overnight mail (p = 0.04), including a certificate of a appreciation (p = 0.05) and later delivery (p = 0.09) improved response rates.

One SWAT [35] conducted a meta-analysis and found no evidence of a difference in retention rates when a festive greetings card was used compared to when it was not (Odds ratio: 0.96, 95% CI 0.71 to 1.79, p = 0.77).

Intervention cost was only reported by one of the included studies [35], however cost effectiveness was not assessed due to primary outcome finding no evidence of additional retention. The impact of the intervention on retention at subsequent timepoints was not assessed by any of the included studies.

A further three retention SWATs [36, 38, 39] were identified during the search but remained ongoing, with no data reported, at the time of analysis. Two studies [36, 39] were hosted in a surgical trial, and focused on postal vs telephone follow up and inclusion of a theoretically informed questionnaire cover letter vs a generic letter respectively. The remaining study by Arundel et al. [38] was hosted in a wound care trial and focused again on a thank you card sent between follow up timepoints.

## Discussion

This review identified 13 eligible randomised controlled studies within a trial of recruitment and retention interventions for surgical or wound care studies. Due to the heterogeneity between interventions, it was not possible to combine any studies in a meta-analysis.

The majority of recruitment studies focussed on consent provision, which correlates with the findings of the Cochrane review by Treweek et al. [11] where modification to consent processes or the methods by which information was presented were the most frequent SWATs. Only one study included in this review [23] relating to consent materials reported a statistically significant effect of the intervention that the enhanced consent materials used reduced the odds of consent. Only one other study found a statistically significant benefit to recruitment through inclusion of an enhanced TPI intervention [34]. Due to the heterogeneity of included recruitment interventions and the small sample sizes of the existing studies which limits the provision of reliable evidence and ascertainment of intervention effectiveness, additional replications are recommended to build this evidence base and to ascertain GRADE certainty evidence for interventions. None of the recruitment SWATs identified within this review were those identified as priorities by the Cochrane review of recruitment methods for RCTs [11]. Trialists should therefore also consider replication of these priority SWATs in surgical and wound care trials in order to help build the evidence base for these interventions.

The reporting of demographic data in recruitment SWATs identified was poor. Given that many under-served groups are often not represented in trials it should be a key aspect of reporting for recruitment SWATs to ensure that certain populations are not disadvantaged by a recruitment or retention strategy [43, 44]. This also limits the generalisability of these results as those that did report demographic data predominantly included older Caucasian participants.

The majority of the SWATs included in this review focussed on retention in relation to postal questionnaire response rates. This correlates with the recent Cochrane review by Gillies

et al. [10]. Only two of the retention SWATs included in this review [29, 31] identified statistically significant differences in response rates when a pen was included [29] and overnight mail, a certificate of appreciation and later delivery [31] were used. One SWAT [35] found that there was no evidence of a difference in retention rates when a festive greetings card was used compared to when it was not (Odds ratio: 0.96, 95% CI 0.71 to 1.79, p = 0.77) and recommended festive greetings cards should not be used as a method of retention. While the associated meta-analysis includes three studies outside of the eligible patient group for this review, we suggest that this finding still holds for surgical and wound care populations.

As with the recruitment SWATs, additional replications are recommended due to the limited evidence available currently. Three of the retention SWATs [22, 24, 26] identified within this review corresponded to Priority A SWATs (low certainty evidence requiring rigorous replication) identified in the Cochrane review of retention methods for RCTs [10]. These SWATs will contribute to the building evidence base for these interventions however further replications are likely to still be necessary to ensure high certainty evidence of effectiveness is ascertained.

In this review, cost effectiveness of interventions was to be assessed, however this was only reported in two recruitment studies [25, 26] and costs were also reported in one retention study [35]. Findings were to be expected given the associated resource implications; additional visits to sites and recruitment by a urologist rather than a nurse were more expensive. Retention at subsequent time points was not assessed in any of the included retention SWATs. When considering the need for further replications of SWATs, recent guidance [18] indicates that consideration should be given the generalisability of the populations and host trial interventions already included in a meta-analysis of a SWAT intervention. This is an important point to consider in the context of research waste. For example with the inclusion of pens with a postal questionnaire to improve retention, there is only one SWAT in a surgical and wound care population, however there is an existing meta-analysis [45] of pen SWATs across populations which indicates a 1.9% increase in retention when a pen is included with a questionnaire for which there is moderate GRADE certainty overall and high GRADE certainty evidence for older populations. In light of this further replications may not be justified within surgical or wound care populations specifically.

Half of the studies included used clinic follow-up either alone or in conjunction with remote, postal follow-up. Improving attendance at face-to-face visits in trials was an evidence gap identified by the most recent systematic review of retention strategies across all studies [10]. Surgical and wound care trials may therefore be an ideal context in which to test further strategies aimed at enhancing face-to-face follow-up.

## Limitations

Firstly, the majority of studies identified in this review were in relation to surgical rather than wound care trials which limits the applicability of the review to trials in this area looking for potential interventions to include. Two further wound care studies were identified however results were not yet available, and the authors are aware of further SWATs also being conducted in wound care studies [46]. As a result, the limited evidence base should continue to build here over time, however trialists undertaking wound care studies are encouraged to include a randomised SWAT to allow the evidence base for effective interventions to build in this area.

Secondly, limited information on cost effectiveness and impacts on further retention, was available and hence there remains significant uncertainty around potentially cost-effective interventions at this time.

We acknowledge a limitation of our search strategy in that only publications in the English language were included in the review, thus including potential language bias. Given that no SWATs were identified in the search which were written in languages other than English we view the impacts of this limitation to be limited.

Due to the diversity of specialties and conditions related to surgery and wound care, and the variants in the description of SWATs (e.g., nested, SWAT, study within a trial, embedded) it proved difficult to develop a precise search strategy. The strategy therefore opted for increased sensitivity rather than precision by using the overarching terms Surgery and Wound, along with relevant terms as used in the Cochrane reviews [10, 11] in relation to SWATs, to attempt to ensure all relevant studies were captured. We acknowledge that despite this approach, there is potential for some SWATs to have been missed, however we anticipate minimal impact from this due to the range of databases and resources searched.

Finally, we acknowledge potential inaccuracy in relation to the risk of bias assessment completed, due to the fact that the domains in the Risk of Bias 2 tool [21] do not necessarily fit easily with the SWAT design. Risk of Bias was assessed independently by the two authors to mitigate this as far as possible.

## Conclusion

This review has identified the different interventions which have previously or are currently being tested to improve recruitment and retention in surgical and wound care trials. The included studies within a trial had significant variation in interventions used, and the predominance of SWATs conducted thus far in these two areas are within surgical trials. Further SWATs in wound care studies are therefore recommended. Further replications of SWATs previously undertaken in surgical trials are also recommended to ensure clear evidence and certainty of this in relation to interventions, subject to the need for further replications which should be assessed appropriately in line with existing Trial Forge Guidance for the avoidance of research waste [18].

## Supporting information

**S1 File.**
(DOCX)

**S2 File.**
(DOCX)

## Author Contributions

**Conceptualization:** Catherine Arundel.

**Data curation:** Catherine Arundel, Andrew Mott.

**Formal analysis:** Catherine Arundel, Andrew Mott.

**Investigation:** Catherine Arundel, Andrew Mott.

**Methodology:** Catherine Arundel, Andrew Mott.

**Project administration:** Catherine Arundel, Andrew Mott.

**Writing – original draft:** Catherine Arundel.

**Writing – review & editing:** Catherine Arundel, Andrew Mott.

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
