## [Decision Letter · Decision Letter 0]

1 Feb 2023

PONE-D-22-32540Recruitment and retention strategies in surgical and wound care trials: a systematic reviewPLOS ONE

Dear Dr. Arundel,

Thank you for submitting your manuscript to PLOS ONE. After careful consideration, we feel that it has merit but does not fully meet PLOS ONE’s publication criteria as it currently stands. Therefore, we invite you to submit a revised version of the manuscript that addresses the points raised during the review process.

We look forward to receiving your revised manuscript.

Kind regards,

Elisa Ambrosi

Academic Editor

PLOS ONE

Journal Requirements:

3. We note that this manuscript is a systematic review or meta-analysis; our author guidelines therefore require that you use PRISMA guidance to help improve reporting quality of this type of study. Please upload copies of the completed PRISMA checklist as Supporting Information with a file name “PRISMA checklist”."

Reviewers' comments:

Reviewer's Responses to Questions

**Comments to the Author**

1. Is the manuscript technically sound, and do the data support the conclusions?

Reviewer #1: Yes

Reviewer #2: Yes

2. Has the statistical analysis been performed appropriately and rigorously? 

Reviewer #1: N/A

Reviewer #2: N/A

3. Have the authors made all data underlying the findings in their manuscript fully available?

Reviewer #1: Yes

Reviewer #2: No

4. Is the manuscript presented in an intelligible fashion and written in standard English?

Reviewer #1: Yes

Reviewer #2: Yes

5. Review Comments to the Author

Reviewer #1: The article is well written and addresses a substantial problem for researchers: patient enrolment and retention.

The abstract is well written.

The introduction introduces appropriately the reader to the topic. However, it doesn't appear clear why the authors decided to investigate both surgical trials and wound trials. Please argument why the two populations can be analysed together.

Discussion "findings were not unsurprising; additional

visits to sites and recruitment by a urologist rather than a nurse were more expensive.

Retention at subsequent time points was not assessed in any of the included retention

SWATs." In this paragraph I would expect the authors to justify why they find the results not unsurprising. Please, consider also revise the terms "not unsurprising2, which may be difficult to understand for someone not speaking in English as first language.

Moreover, it would be nice to comment the replicability of the results in the discussion in relation to the limited part of the population described (white, males).

Reviewer #2: Thank you for the opportunity to review this manuscript which presents the findings from a systematic review of interventions to improve recruitment and/or retention in clinical trials set within surgery or wound care. I have some specific points for consideration which are listed below.

Abstract

• For both the abstract and the introduction I think the rationale for the need for this review could be clearer. As written it states that recruitment and retention are an ongoing issue for trials. There are lots of operations and people living with chronic wounds and therefore assessment of this context is important. I would argue there needs to be a more direct link between recruitment and retention and surgical and wound care trials, rather than high numbers in clinical care. For example, do they routinely under-recruit/retain – what is the problem that this review is addressing? This needs strengthened.

• Within abstract methods section helpful to state ‘within a surgical or wound based host randomised controlled trial’ so as to be consistent with terminology throughout re host trials – which will help readers.

• There isn’t any information on the analysis methods used in the methods section.

• Results section states 11 studies were included but in the main text it states 12 and 12 studies are included in the tables.

Introduction

• Within the opening sentences there are several uses of ‘their’. Sometimes it isn’t clear which ‘their’ the author is referring to. Maybe clearer to be more specific e.g sentence 2 – The validity and reliability of RCTs is highly dependent on recruiting and retaining….’

• Within introduction sometimes the ‘interventions’ are referred to as methods sometimes strategies – would be helpful to be consistent.

• Point re rationale made above for abstract also needs addressed here.

• I didn’t understand the second section of the last sentence of the second paragraph.

• Last sentence of paragraph 3. The point being made re evidence from quasi or non-randomised designs – is this in relation to the design of the recruitment and retention trial or the host trial? I didn’t then understand how this links to the point re ‘real’ trials. Isnt that more that some trials included in the Cochrane recruitment review were hypothetical?

Methods

• Sentence that states studies were not eligible if they were hypothetical needs clarification. If the host parent trials was hypothetical? The SWAT? Both?

• Were any other types of studies excluded?

• Would be helpful to state how the outcomes of recruitment and retention were defined.

• Was unit of randomisation collected as a SWAT characteristic?

• In Synthesis section it states a flowchart ‘will be presented’ – needs changed to ‘are’

Results

• Reports that 12 studies were included – ensure address mismatch with abstract

• When stating studies use a direct approach – can you give an i.e. to hep the reader.

• Likewise when stating postal follow up – maybe include postal questionnaire follow up.

• The results state that one intervention (patient information video decision aid) was evaluated in ‘multiple’ studies. It was evaluated in 2 studies – why weren’t these studies considered for meta-analysis?

• Risk of bias – states 11 studies were included.

• Recruitment section

o second para opens stating 4 of the 6 SWATs – but weren’t there 7? 4 on consent info, 1 on study set up and 2 on training.

o I think it would be helpful to provide more information on ‘consent information’ interventions. This could mean the content and/or the mode pf delivery of information has been modified. Would be helpful to provide further details – could consider using similar categories to the Cochrane review.

o Para 7 – should be ‘were’ not ‘was’ identified.

• Retention section

o Last sentence of first para – were these host trials that were factorial?

o 2nd para last sentence – I wasn’t clear why the authors report the SWATs focused on provision of consent information and study set up – isn’t that relevant for the recruitment interventions not retention.

• Could the authors comment on why the following paper wasn’t included as some of the included trials are surgical–

o Coleman E, Arundel C, Clark L, Doherty L, Gillies K, Hewitt C et al. Bah humbug! Association between sending Christmas cards to trial participants and trial retention: randomised study within a trial conducted simultaneously across eight host trials BMJ 2021; 375 :e067742

Discussion

• Discussion needs edited base don ‘consent information; interventions description being expanded

• 2nd para last sentence – reads that replications are required due to heterogeneity of interventions. But replications are also required due to small sample sizes which means that individual studies don’t provide high quality, reliable, evidence. This point needs to be covered in discussion.

• Would be helpful to also cover priorities for recruitment and retention intervention testing as stated in Cochrane reviews.

6. PLOS authors have the option to publish the peer review history of their article (what does this mean?). If published, this will include your full peer review and any attached files.

Reviewer #1: No

Reviewer #2: **Yes: **Dr Katie Gillies

---

## [Author Response · Author response to Decision Letter 0]

6 Mar 2023

Reviewer 1

The introduction introduces appropriately the reader to the topic. However, it doesn't appear clear why the authors decided to investigate both surgical trials and wound trials. Please argument why the two populations can be analysed together.

 We are pleased that the introduction provided a helpful summary of the topic but apologise it was not clear why the populations were combined. We have updated the introduction section to reflect that surgery often leads to wounds and therefore the populations are likely to be similar, hence the grouping here.

Discussion "findings were not unsurprising; additional visits to sites and recruitment by a urologist rather than a nurse were more expensive. Retention at subsequent time points was not assessed in any of the included retention SWATs." 

In this paragraph I would expect the authors to justify why they find the results not unsurprising. Please, consider also revise the terms "not unsurprising”, which may be difficult to understand for someone not speaking in English as first language.

 We apologise that this wording was ambiguous. The Discussion has been updated accordingly to reflect the results were expected given the resource implications associated with the expensive interventions.

Moreover, it would be nice to comment the replicability of the results in the discussion in relation to the limited part of the population described (white, males).

 This suggestion to update the discussion to reflect the EDI of participants has been actioned accordingly to reflect the population is predominantly older patients, largely white ethnicity.

Reviewer 2

For both the abstract and the introduction I think the rationale for the need for this review could be clearer. As written it states that recruitment and retention are an ongoing issue for trials. There are lots of operations and people living with chronic wounds and therefore assessment of this context is important. I would argue there needs to be a more direct link between recruitment and retention and surgical and wound care trials, rather than high numbers in clinical care. For example, do they routinely under-recruit/retain – what is the problem that this review is addressing? This needs strengthened.

 We are sorry that the rationale for the review was not clear. We have updated the abstract and introduction to make clear the difficulties with recruitment/retention in surgical trials and given that surgery often leads to wounds and therefore the populations are likely to be similar, hence why these were grouped together.

Within abstract methods section helpful to state ‘within a surgical or wound based host randomised controlled trial’ so as to be consistent with terminology throughout re host trials – which will help readers.

 We thank the reviewer for this note re consistency and have updated the Abstract – Methods accordingly.

There isn’t any information on the analysis methods used in the methods section.

 We thank the reviewer for noting this missingness. The Abstract – Methods has been updated to reflect the analysis methods used.

Results section states 11 studies were included but in the main text it states 12 and 12 studies are included in the tables.

 We apologise for the inaccuracy of the reporting here. This was clearly missed when we updated the manuscript with an updated search. The Abstract – Results section has been updated accordingly.

Within the opening sentences there are several uses of ‘their’. Sometimes it isn’t clear which ‘their’ the author is referring to. Maybe clearer to be more specific e.g sentence 2 – The validity and reliability of RCTs is highly dependent on recruiting and retaining….’

 We appreciate that the use of their may have caused ambiguity in the initial sentences of the Introduction and have amended this accordingly for clarity.

Within introduction sometimes the ‘interventions’ are referred to as methods sometimes strategies – would be helpful to be consistent.

 We thank the reviewer for their suggestion here – the manuscript has been reviewed and updated throughout for consistency

Point re rationale made above for abstract also needs addressed here. As above we have updated the abstract and introduction to better reflect the rationale.

I didn’t understand the second section of the last sentence of the second paragraph.

 We apologise this point was not clear. On review this duplicates information provided in Paragraph 3 and so has been removed

Last sentence of paragraph 3. The point being made re evidence from quasi or non-randomised designs – is this in relation to the design of the recruitment and retention trial or the host trial? I didn’t then understand how this links to the point re ‘real’ trials. Isnt that more that some trials included in the Cochrane recruitment review were hypothetical? We have made clearer that the quasi and non randomised designs are relating to the SWAT and have also made reference to hypothetical designs as suggested. We have removed the reference to ‘real’ trials for avoidance of confusion

Sentence that states studies were not eligible if they were hypothetical needs clarification. If the host parent trials was hypothetical? The SWAT? Both?

 The Methods – Eligibility Criteria section has been clarified as requested.

Were any other types of studies excluded?

 The inclusion criteria note studies as eligible for inclusion if the host trial and SWAT were randomised. We have amended the exclusion criteria to make clear non randomised or quasi randomised designs were not eligible to make this clearer.

Would be helpful to state how the outcomes of recruitment and 

retention were defined.

 As requested definitions of recruitment and retention have been added to the Methods – Outcome data section

Was unit of randomisation collected as a SWAT characteristic?

 We did not separately extract unit of randomisation but this was collected as part of the number of participants recruited to each arm, whereby this was noted as the number of sites for relevant studies. We have updated the Methods – Data Items to include this.

In Synthesis section it states a flowchart ‘will be presented’ – needs changed to ‘are’

 We apologise for the oversight in tense here. This has been updated accordingly.

Reports that 12 studies were included – ensure address mismatch with abstract

 We apologise for the inaccuracy of the reporting here. This was clearly missed when we updated the manuscript with an updated search. The Abstract – Results section has been updated accordingly.

When stating studies use a direct approach – can you give an i.e. to hep the reader.

 This is a useful addition and we have added as suggested to the Results – Study Characteristics section.

Likewise when stating postal follow up – maybe include postal questionnaire follow up.

 This is a useful addition and we have added as suggested to the Results – Study Characteristics section.

The results state that one intervention (patient information video decision aid) was evaluated in ‘multiple’ studies. It was evaluated in 2 studies – why weren’t these studies considered for meta-analysis?

 The authors have re-reviewed this decision and agree that the two interventions were too heterogeneous to warrant combining via meta analysis. We have updated the results section to remove any ambiguity here.

Risk of bias – states 11 studies were included We apologise for the inaccuracy of the reporting here. This was clearly missed when we updated the manuscript with an updated search. The Results – Risk of Bias section has been updated accordingly. In undertaking this we noted that the assessment for Parker et al was inadvertently missed from the reporting – this has been added accordingly.

Recruitment section

o second para opens stating 4 of the 6 SWATs – but weren’t there 7? 4 on consent info, 1 on study set up and 2 on training.

o I think it would be helpful to provide more information on ‘consent information’ interventions. This could mean the content and/or the mode pf delivery of information has been modified. Would be helpful to provide further details – could consider using similar categories to the Cochrane review.

o Para 7 – should be ‘were’ not ‘was’ identified. We apologise for the inaccuracy of the reporting here. This was clearly missed when we updated the manuscript with an updated search. The Results – Recruitment section has been updated accordingly.

This is a helpful suggestion and we have updated the manuscript to make clear the types of consent information interventions.

We apologise for the oversight in tense here. This has been updated accordingly.

Retention section

o Last sentence of first para – were these host trials that were factorial?

o 2nd para last sentence – I wasn’t clear why the authors report the SWATs focused on provision of consent information and study set up – isn’t that relevant for the recruitment interventions not retention.

It was the SWATs which were factorial – we have updated the Results – Retention section for clarity

We thank the reviewer for spotting this oversight- you are correct the provision of consent information and set up should not be detailed here. We have revised this section to report the retention interventions correctly.

Could the authors comment on why the following paper wasn’t included as some of the included trials are surgical–

o Coleman E, Arundel C, Clark L, Doherty L, Gillies K, Hewitt C et al. Bah humbug! Association between sending Christmas cards to trial participants and trial retention: randomised study within a trial conducted simultaneously across eight host trials BMJ 2021; 375 :e067742

 At the time of initial search this publication was not available so not included and it appears the subsequent search did not pick up this publication either, possibly due to the timing of indexing. We acknowledge that this should have been included however and so have added this in accordingly. 

Discussion needs edited based on ‘consent information; interventions description being expanded

 We have updated the Discussion section to reflect the types of consent information interventions included.

2nd para last sentence – reads that replications are required due to heterogeneity of interventions. But replications are also required due to small sample sizes which means that individual studies don’t provide high quality, reliable, evidence. This point needs to be covered in discussion

 This is an excellent point – we have updated this section of the Discussion to reflect the impacts of the sample sizes here.

Would be helpful to also cover priorities for recruitment and retention intervention testing as stated in Cochrane reviews. This is a helpful suggestion and one which has now been reflected on within the Discussion section

---

## [Decision Letter · Decision Letter 1]

26 Apr 2023

PONE-D-22-32540R1Recruitment and retention interventions in surgical and wound care trials: a systematic reviewPLOS ONE

Dear Dr. Arundel,

Thank you for submitting your manuscript to PLOS ONE. After careful consideration, we feel that it has merit but does not fully meet PLOS ONE’s publication criteria as it currently stands. Therefore, we invite you to submit a revised version of the manuscript that addresses the points raised during the review process.

We look forward to receiving your revised manuscript.

Kind regards,

Elisa Ambrosi

Academic Editor

PLOS ONE

Reviewers' comments:

Reviewer's Responses to Questions

**Comments to the Author**

1. If the authors have adequately addressed your comments raised in a previous round of review and you feel that this manuscript is now acceptable for publication, you may indicate that here to bypass the “Comments to the Author” section, enter your conflict of interest statement in the “Confidential to Editor” section, and submit your "Accept" recommendation.

Reviewer #1: (No Response)

Reviewer #2: All comments have been addressed

2. Is the manuscript technically sound, and do the data support the conclusions?

Reviewer #1: Partly

Reviewer #2: Yes

3. Has the statistical analysis been performed appropriately and rigorously? 

Reviewer #1: N/A

Reviewer #2: N/A

4. Have the authors made all data underlying the findings in their manuscript fully available?

Reviewer #1: Yes

Reviewer #2: Yes

5. Is the manuscript presented in an intelligible fashion and written in standard English?

Reviewer #1: Yes

Reviewer #2: Yes

6. Review Comments to the Author

Reviewer #1: Abstract: the introduction is not clear and it doesn't introduce properly the two populations under scrutiny and why retention is a problem. In the abstract the authors state there are 7 articles included for recruitment, but then they say "four out of the SIX".

Introduction: as the previous reviewer undelined, the authors didn't justify clearly why surgical and wound care trials against all other trials, need to be addressed in terms of retention strategies, there needs to be a more direct link between recruitment and retention and surgical and wound care trials, rather than high numbers in clinical care. Moreover, in the comments to the authors, the authors declare that they included 12 articles, but in reality in the manuscript there are 13 articles.

Analysis: if there are 13 RCTs, it should be justified why the authors talk about 14 interventions.

Results ". A further record was subsequently identified for inclusion given 62.5% of included studies were surgical or wound care related". This sentence is not clear to me: Why this further record has not been included among the 13? why is it relevant to state that 62.5% of the studies where surgical or wound care related?

Page 24: please consider changing "results where not unsurprising". Discussion: in the last paragraph the authors state that "Half of the studies included used clinic follow-up either alone or in conjunction with remote

follow-up. It would be interesting to discuss what are the forms of remote follow up. A recent systematic review by Mette Brøgger-Mikkelsen et al (2020) hightlighted that Online recruitment was both superior in regard to time efficiency and cost-effectiveness compared with offline recruitment and this could be an effective strategy to improve the low attendance at face-to-face visits in trials.

Moreover, the systematic review cited by the authors in the last paragraph needs to be correctly cited.

Reviewer #2: (No Response)

7. PLOS authors have the option to publish the peer review history of their article (what does this mean?). If published, this will include your full peer review and any attached files.

Reviewer #1: No

Reviewer #2: **Yes: **Dr Katie Gillies

---

## [Author Response · Author response to Decision Letter 1]

15 May 2023

Abstract: the introduction is not clear and it doesn't introduce properly the two populations under scrutiny and why retention is a problem. 

We apologise that the introduction was unclear as to the two populations under scrutiny. We have sought to revise the introduction section to make this clearer whilst remaining within the confines of the journal word count for the abstract.

In the abstract the authors state there are 7 articles included for recruitment, but then they say "four out of the SIX". We apologise for the oversight here when revising this manuscript. This has been corrected accordingly.

Introduction: as the previous reviewer undelined, the authors didn't justify clearly why surgical and wound care trials against all other trials, need to be addressed in terms of retention strategies, there needs to be a more direct link between recruitment and retention and surgical and wound care trials, rather than high numbers in clinical care. 

Moreover, in the comments to the authors, the authors declare that they included 12 articles, but in reality in the manuscript there are 13 articles.

We are sorry that the amendments made to the introduction did not satisfactorily deal with the previous reviewer comments. We have made further revisions to the introduction to note the difficulties in both recruitment and retention in these trial types.

We have thoroughly checked the manuscript to ensure that it is clear that 13 articles in total were included in the review.

Analysis: if there are 13 RCTs, it should be justified why the authors talk about 14 interventions.

We apologise for the inaccuracy here. The analysis section has been amended to remove details of the number of interventions for avoidance of any confusion.

A further record was subsequently identified for inclusion given 62.5% of included studies were surgical or wound care related". This sentence is not clear to me: Why this further record has not been included among the 13? why is it relevant to state that 62.5% of the studies where surgical or wound care related? 

We apologise for the oversight here when revising the manuscript. The original number of studies included should have been reported as 12, with the additional record increasing this to 13. This has been updated accordingly for clarity.

We suggest it is relevant to report the proportion of surgical/wound care studies in the final included record, given this is a simultaneous SWAT (i.e., conducted in multiple studies at the same time) and not all the studies included would meet the inclusion criteria for this review.

Page 24: please consider changing "results where not unsurprising". 

We thank the reviewer for this comment. We have re reviewed the manuscript and amended this sentence in relation to the cost effectiveness of recruitment strategies.

Discussion: in the last paragraph the authors state that "Half of the studies included used clinic follow-up either alone or in conjunction with remote follow-up. It would be interesting to discuss what are the forms of remote follow up. A recent systematic review by Mette Brøgger-Mikkelsen et al (2020) hightlighted that Online recruitment was both superior in regard to time efficiency and cost-effectiveness compared with offline recruitment and this could be an effective strategy to improve the low attendance at face-to-face visits in trials.

Moreover, the systematic review cited by the authors in the last paragraph needs to be correctly cited. 

We have added clarification to this sentence to reflect that the remote follow up methods reported were postal. We thank the reviewer for flagging the review by Mette Brogger Mikkelsen et al but have not commented further on this given neither online recruitment nor follow up methods were used by any of the included trials.

We apologise for the incorrect placing of this citation. This has been corrected accordingly.

---

## [Decision Letter · Decision Letter 2]

19 Jun 2023

Recruitment and retention interventions in surgical and wound care trials: a systematic review

PONE-D-22-32540R2

Dear Dr. Arundel,

We’re pleased to inform you that your manuscript has been judged scientifically suitable for publication and will be formally accepted for publication once it meets all outstanding technical requirements.

Kind regards,

Elisa Ambrosi

Academic Editor

PLOS ONE

Additional Editor Comments (optional):

Reviewers' comments:

Reviewer's Responses to Questions

**Comments to the Author**

1. If the authors have adequately addressed your comments raised in a previous round of review and you feel that this manuscript is now acceptable for publication, you may indicate that here to bypass the “Comments to the Author” section, enter your conflict of interest statement in the “Confidential to Editor” section, and submit your "Accept" recommendation.

Reviewer #1: All comments have been addressed

Reviewer #2: All comments have been addressed

2. Is the manuscript technically sound, and do the data support the conclusions?

Reviewer #1: Yes

Reviewer #2: Yes

3. Has the statistical analysis been performed appropriately and rigorously? 

Reviewer #1: Yes

Reviewer #2: N/A

4. Have the authors made all data underlying the findings in their manuscript fully available?

Reviewer #1: Yes

Reviewer #2: No

5. Is the manuscript presented in an intelligible fashion and written in standard English?

Reviewer #1: Yes

Reviewer #2: Yes

6. Review Comments to the Author

Reviewer #1: The authors have addressed all previous concerns. The number of the articles has been fixed and the additional comments revised.

Reviewer #2: (No Response)

7. PLOS authors have the option to publish the peer review history of their article (what does this mean?). If published, this will include your full peer review and any attached files.

Reviewer #1: No

Reviewer #2: **Yes: **Katie Gillies

---

## [Editor Report · Acceptance letter]

12 Jul 2023

PONE-D-22-32540R2 

Recruitment and retention interventions in surgical and wound care trials: a systematic review 

Dear Dr. Arundel:

I'm pleased to inform you that your manuscript has been deemed suitable for publication in PLOS ONE. Congratulations! Your manuscript is now with our production department. 

Kind regards, 

on behalf of

Dr. Elisa Ambrosi 

Academic Editor

PLOS ONE